# Chronic Mild Unpredictable Stress and High-Fat Diet Given during Adolescence Impact Both Cognitive and Noncognitive Behaviors in Young Adult Mice

**DOI:** 10.3390/brainsci11020260

**Published:** 2021-02-19

**Authors:** Stephen L. P. Lippi

**Affiliations:** Department of Psychology & Sociology, Angelo State University, ASU Station #10907, San Angelo, TX 76909, USA; stephen.lippi@angelo.edu; Tel.: +1-325-486-6923

**Keywords:** diet, stress, mice, learning, memory, behavior

## Abstract

Stress and diet are intricately linked, and they often interact in a negative fashion. Increases in stress can lead to poor food choices; adolescence is a period that is often accompanied by increased levels of stress. Stress and poor dietary choices can affect learning and memory; it is important to understand their combined effects when occurring during crucial developmental periods. Here, we present evidence that chronic mild unpredictable stress (CMUS) and high-fat diet (HFD) impact both cognitive and noncognitive behaviors when assessed after four weeks of manipulation in four-week old mice. CMUS mice had increased anxiety in the open field test (OFT) (*p* = 0.01) and spent more time in the open arms of the elevated zero maze (EZM) (*p* < 0.01). HFD administration was shown to interact with CMUS to impair spatial memory in the Morris Water Maze (MWM) (*p* < 0.05). Stress and diet also led to disturbances in non-cognitive behaviors: CMUS led to significantly more burrowing (*p* < 0.05) and HFD administration led to the poorer nest construction (*p* < 0.05). These findings allow for researchers to assess how modifying lifestyle factors (including diet and stress) during adolescence can serve as a potential strategy to improve cognition in young adulthood.

## 1. Introduction

The period of adolescence is an important time of development involving both physical and neurological changes. This period of time involves the maturation of neural connections, leading to changes in cognitive functions and brain structures including both gray and white matter [1]. During the period of adolescence, numerous factors, including stress [2,3,4,5] and diet and exercise [6,7,8,9], can influence physical and cognitive development. These factors are lifestyle factors that can be modified to lead to improved health. Understanding how these factors can both positively and negatively impact health can allow researchers to study and propose ways to improve adolescent health. This may ultimately lead to a healthier transition into adulthood.

The food that is brought into the body has a substantial effect on not only physical/cardiovascular health [10], but also cognitive health [11,12,13]. In fact, the National Center for Chronic Disease Prevention and Health Promotion [14] lists poor nutrition as a risk factor for preventable chronic diseases. School students, from grades 2 through 11, have identified the major benefits of consuming healthy foods to include helping cognitive performance, feeling good physically, and psychological benefits [15]. Although positive effects of consuming healthy food were identified, students in this study [15] also remarked on barriers that are associated with eating healthy, including: convenience of other non-healthy foods, social reinforcement, and simply having a preference for less healthy alternatives to healthy foods. O’Dea et al. (2003) [15] showed that students are aware of the benefits that eating healthy can have; however, there are many factors that may pull them away from eating healthy.

The period of adolescence has shown a high prevalence of obesity [16,17], and obesity in mice has been shown to impact memory, both in behavioral tests [7] and through epigenetic modifications [18]. Western diets and poor nutrition have been studied in the literature and they show that components, such as saturated fats and simple carbohydrates (ex: simple sugars—glucose), can have negative effects on cognition [13,19,20], as well as physical health, including contributing to obesity, heart disease, and diabetes/insulin resistance [21,22,23].

The adolescent brain is vulnerable to stress, and this can have negative consequences on brain development [24]. Stress is felt when we experience some difficulty or adverse external stimulus. The brain reacts to the presence of an external stressor by the activation of the hypothalamic-pituitary-adrenal (HPA) axis, which helps to secrete stress hormones; once the stressor is gone, the system begins to return to baseline through a series of feedback loops [25]. If it is unable to be dealt with, stress can become chronic; chronic stress can often lead to negative health outcomes, including obesity [26,27,28], increased risk of becoming ill [29], and heart disease [30].

Stress has particularly strong negative effects on memory, as noted by the hippocampus’ sensitivity to stress [31]. Numerous chronic stress paradigms [32,33,34,35,36] have been used in animal models to assess the changes in various behaviors and physiology as a result of increased stress. Behaviors that have been assessed as a result of chronic stress include wheel running activity [35], hippocampal-dependent spatial memory [34], and anxiety [5,37]. Chronic stress has been shown to increase both cognitive impairment and plaque deposition in the APP/PS1 mouse model of AD [34], induce anhedonia and impairments in spatial memory in young mice [38], and reduce the hippocampal dentate gyrus volume [39]. Because both stress and poor dietary choices can lead to disruptions in behavior and neurological health, one must consider their combined effects, as so often occurs in the period of adolescence.

As noted above, poor dietary choices can have impacts on not only behavior, but also neuronal markers. Stress also affects adolescents; however, what are the effects of both of these factors on this crucial stage of development? Cartwright et al. (2003) [40] examined the connection between stress and diet in a large sample of adolescents, and found that higher perceived stress levels were related to higher odds of consuming more fatty foods than the least stressed group. Few studies have examined both chronic stress and intake of high-fat diets on behavior; in those that have, often only one sex has been used and diet administration starts at varying time points [41,42]. Kuo et al. (2008) [41] demonstrated neuropeptide Y’s involvement using in vitro and in vivo methodology; however, no behavior results were reported. Another study assessed changes in behavior as a result of HFD and chronic stress [42]; however, these authors used only male rats and started administration of diets after 9 weeks of age. Accordingly, while there are experiments combining these lifestyle factors, many focus on either specific tests of memory or focus on just the physiological effects.

Chronic unpredictable mild stress with HFD administration has also been shown to affect the gut microbiota of mice [36] and lipid metabolism, with the levels of total cholesterol and LDL being higher than in control animals [43]. Additionally, both chronic stress and HFD administration have shown effects on the cardiovascular system [44], metabolic and insulin systems [32,34], and alteration of levels of crucial hunger peptides [41]. Experiments of this nature (manipulating stress and dietary intake) that measure both cognitive and noncognitive behaviors allow for researchers to draw conclusions regarding how an adolescent can be impacted if these lifestyle factors are not positively adjusted.

In the present study, we assessed chronic mild unpredictable stress (CMUS) and HFD effects when given to mice during 4–8 weeks; this was done in both male and female mice, given that sex differences have been observed in mice undergoing diet-induced obesity [45]. Behavioral tests were run assessing anxiety, spatial memory, and noncognitive activities of daily living (ADLs) in order to investigate what effects HFD and CMUS would have on behavior in the early adult mouse after sustaining these factors during the crucial developmental time of adolescence. The results showed that stressed mice exhibited behavioral disinhibition, impaired spatial memory, and built better nests than non-stressed (NS) mice. Mice that were given a HFD had impaired nesting scores and, at several times, there was an interaction between the diet and stress.

## 2. Materials and Methods

### 2.1. Methods

#### 2.1.1. Animals

C57BL/6J mice (Stock No: 000664) were ordered from the Jackson Laboratory (Jax—Bar Harbor, ME, USA). Male and female mice were ordered for breeding purposes; offspring were generated in house. Offspring were weaned and ear-punched for identification purposes at postnatal day (PND) 21. The mice were housed with littermates based on experimental group and sex (Table 1). All of the animals used in this study were of the F1 generation. The animal colony was maintained on a 12 h light/dark cycle (“lights-on” occurred at 08:00 h).

#### 2.1.2. Diets

Diets were purchased through Research Diets Inc. (New Brunswick, NJ, USA). Animals received either a high-fat diet (HFD) (D12492 Rodent Diet with 60 kcal% fat) or a control diet (CD) (D12450J Rodent Diet with 10 kcal% fat; matching sucrose to D12492) starting at PND 28 and continued until the completion of the study. The animals were weighed weekly throughout the progression of the study to keep track of weight changes. Food intake was also measured throughout the duration of the study; particularly, during the four weeks of CMUS administration. However, due to group housing, average intake per mouse was calculated as: [((g) food consumed after one week)/Number of animals in the cage]. The animals received ad libitum access to food and water.

#### 2.1.3. Chronic Mild Unpredictable Stress (CMUS) Paradigm

A CMUS paradigm was administered to animals at four weeks of age (28 days); stressors were adapted and modeled from other CMUS protocols [33,34,35] and their order was randomly assigned across the four weeks of stress treatment. The mice received four days of stressors each week, with each day involving two stressors. The two stressors were randomly given with one occurring in the first half of the light cycle and the second occurring in the second half of the light cycle. Table 2 details the stressors and their procedures.

#### 2.1.4. Behavioral Tests

All of the behavioral tests were done during the “lights-on” period (08:00 h–20:00 h).

##### Open Field Test (OFT)

The OFT is a measure of exploratory behavior and general activity, and it is commonly used as a “control” assay for other behavioral tests that involve activity (such as the Elevated Zero Maze or Morris Water Maze). The apparatus used was one white box measuring 45 × 45 × 40 cm that was constructed from material to prevent absorbing various odors (Harvard Apparatus). The OFT container was divided into a center and a surround for coding purposes. Animals were gently placed facing a back wall (in the surround) and activity was recorded over a 5-min. period using the SMART video tracking system (Panlab, Harvard Apparatus, Holliston, MA, USA). Percent time spent in the center, number of entries into the center region, latency to enter into the center region, and distance travelled were measured. After each animal completed its trial, the animal was removed, and the OFT apparatus was cleaned with 70% ethanol to eliminate any potential odor cues between mice.

##### Elevated-Zero Maze (EZM)

The EZM is a test that measures anxiety and risk-taking behavior. The apparatus is an “O” shaped platform that is raised above the ground and it is divided into two closed sections with walls surrounding the edges and two open sections with no surrounding walls. Additionally, the EZM does not have a middle intersection, which forces the mice to be either in an open or a closed section at any given time [46]. Each animal was given a single 5-min. trial that began with each mouse being placed in a closed portion of the maze facing inward; behavior was tracked using a ceiling-mounted camera that was positioned over the maze (Harvard Apparatus). An animal was considered in a given arm when all four paws were within that section. Time spent in each section of the maze (open v. closed) and the total number of entries into each section were recorded. After each animal completed its trial, the animal was removed and the EZM was cleaned with 70% ethanol to eliminate any potential odor cues between mice.

##### Morris Water Maze (MWM)

The MWM is used to assess spatial memory in rodents [47]. The MWM tub (four-foot diameter) was filled with opaque water (water made opaque through the addition of non-toxic white paint) to keep the animals from seeing the platform and it was surrounded by a white curtain (approximately 12 inches away from the tub). Visual cues (black and white images) were located around the tub, serving as spatial cues. Each animal was placed gently in the tub facing the edge of the tub. Start locations were consistent for all mice each day; the start locations varied between days. Each animal was given 60 s (s) to find the hidden platform while being tracked by an overhead camera (using the SMART Video tracking system (Panlab), Harvard Apparatus). The latency to find the platform, percent time spent in the target quadrant, and number of platform crosses on probe trials were measured. In addition, thigmotaxicity (time spent in the outer edges of the pool) was measured. At the end of 60 s, if the mouse failed to find the platform, the animal was gently guided to the platform where it remained for 10 s before being removed. Upon completion of the trial, the animal was dried with a towel and then placed under a heating lamp for 45 s before beginning the next trial. Once the animal completed all three trials, it was finished for the day. On days 2, 4, and 6, trial 3 was a probe trial (platform was lowered, so it became inaccessible during the 60 s trial). Day 7 consisted of only one trial, which was a probe trial. This trial was used to assess long term memory of the platform’s location.

##### Activities of Daily Living (ADLs): Burrowing

Burrowing is a common behavior in rodents that provides shelter, but also allows for defense against larger predators and storage of food [48]. The mice were individually housed to measure burrowing behavior. Three hours before the onset of the dark cycle, mice were placed in individual shoebox cages (Ancare) with bedding and a PVC pipe filled with 250 g of pea-gravel (small rocks). Two measurements were taken during the burrowing assay: the amount of pea-gravel burrowed after 2 h and amount burrowed overnight. The two-hour measurement is a more sensitive measure, since this is still occurring in the light phase, as opposed to the dark phase when activity is increased [49]. The following morning, the remaining pea-gravel was weighed to obtain an overnight measurement. The shoebox cages were then emptied and prepped for the nesting assay (see below, Section Activities of Daily Living (ADLs): Nesting).

##### Activities of Daily Living (ADLs): Nesting

Fresh bedding was placed in each individual shoe box cage and 3 g of shredded white paper was scattered in the cage. Shredded paper was used, since nests constructed out of this material have been shown to have higher inter-rater reliability among raters blind to experimental conditions when compared to other materials, including cotton squares or paper twists [50]. After 24 h, the nests were photographed and rated by trained raters blind to experimental condition on a scale of 1–5: 1 represented that no nest was constructed while a 5 represented a full nest had been constructed. Intra-class correlations were calculated to test for reliability between raters. 

### 2.2. Statistical Analyses

Data were analyzed through the use of SPSS v.19 and the graphs were created using GraphPad Prism 8 and the *ggplot2* package in R [51,52]. Mixed ANOVAs were run for MWM analysis, while three-way ANOVAs (diet, stress, and sex as factors) were run for dependent variables that were measured in the OFT, EZM, and ADL measures. A Greenhouse–Geisser correction was applied when assumptions of sphericity were violated. All of the plotted data are mean (*M*) ± standard error of the mean (SEM). *p* < 0.05 was designated as statistically significant and *p* < 0.1 indicated a trend. Any main effects were followed up by Bonferroni *post-hoc* analysis. The simple effects analyses were run as follow-ups to significant interactions.

## 3. Results

### 3.1. Animal Weights and Food Consumed

A 2 × 2 × 2 × 7 mixed ANOVA was run for weight over time, including stress, diet, and sex as independent variables and weeks (time) as the within-subject variable. Between-subjects effects were seen for sex, *F*(1, 29) = 108.321, *p* < 0.001, partial η^2^ = 0.789, and diet, *F*(1, 29) = 16.318, *p* < 0.001, partial η^2^ = 0.360, with males weighing significantly more than females and mice on HFD weighing significantly more than those on CD.

There was a significant effect of week, *F*(1.536, 44.557) = 382.417, *p* < 0.001, partial η^2^ = 0.930; throughout the experiment, the weights increased significantly (Figure 1). In addition, there were significant interactions of week*sex, *F*(1.536, 44.557) = 14.197, *p* < 0.001, partial η^2^ = 0.329 and week*stress, *F*(1.536, 44.557) = 33.405, *p* < 0.01, and partial η^2^ = 0.217. Males weighed significantly more than females each week (baseline (PND 28), *p* < 0.01; all other time points, *p* < 0.001). Mice that experienced NS throughout the experiment weighed significantly more than those experiencing CMUS on weeks 2–5 (week 2: *p* < 0.05, week 3: *p* < 0.01, week 4: *p* < 0.01, and week 5: *p* < 0.05) (Figure 2a).

A 2 × 2 × 2 × 4 mixed ANOVA was run for food consumed over the course of the four-week CMUS paradigm, including stress, diet, and sex as independent variables and weeks (time) as the within-subject variable. There were significant interactions of week*diet, *F*(1.827, 52.992) = 9.590, *p* < 0.001, partial η^2^ = 0.249, and week*stress, *F*(1.827, 52.992) = 23.053, *p* < 0.001, partial η^2^ = 0.443. During weeks 3 and 4, mice that were given CMUS ate significantly less than mice given NS: *p* < 0.001 (week 3), *p* = 0.002 (week 4) (Figure 2b).

### 3.2. Open Field Test

#### 3.2.1. Percent Time in the Center

A 2 × 2 × 2 ANOVA was run on percent time spent in the center of the OFT. There was a main effect of stress, *F*(1, 29) = 7.586, *p* = 0.01, partial η^2^ = 0.207. Mice that experienced CMUS spent significantly less time in the center of the OF, which was of increased anxiety. No effects of diet or sex were noted (Figure 3).

#### 3.2.2. Center Entries

A 2 × 2 × 2 ANOVA was run on the number of entries made into the center of the OFT. A significant effect of stress was seen, *F*(1, 29) = 4.382, *p* < 0.05, and partial η^2^ = 0.131. Mice that experienced CMUS made significantly less entries into the center of the OF. No effects of diet or sex were seen (Figure 4).

#### 3.2.3. Latency to Enter the Center

A 2 × 2 × 2 ANOVA was run on the latency to first enter the center of the OFT. No significant effects were seen.

#### 3.2.4. Total Distance

There was a significant diet*sex interaction, *F*(1, 29) = 5.236, *p* < 0.05, and partial η^2^ = 0.153, alongside a significant main effect of sex, *F*(1, 29) = 8.015, *p* < 0.01, and partial η^2^ = 0.217 on the total distance travelled in the OFT. Overall, male mice travelled further than female mice (*p* = 0.008), and males on a CD traveled further than females on a CD, *p* < 0.01. There were no differences between sexes in the HFD condition, with males and females running similar distances (*p* > 0.05). CD males ran further and CD females ran shorter distances on average when compared to their HFD counterparts.

### 3.3. Elevated Zero Maze (EZM)

#### 3.3.1. Percent Time in the Open Arm

A 2 × 2 × 2 ANOVA was run on the percent time in the open arm of the EZM. There was a significant effect of stress, *F*(1, 29) = 9.883, *p* < 0.01, and partial η^2^ = 0.254, and a significant diet*sex interaction, *F*(1, 29) = 4.778, *p* < 0.05, and partial η^2^ = 0.141. Mice that underwent CMUS spent significantly more time in the open arms of the EZM (*p* < 0.01). (Figure 5) Female mice consuming a CD spent significantly more time in the open arms as compared to males on a CD (*p* = 0.01). There was also a trend noted for female mice: those consuming a CD spent more time in the open arms when compared to females on a HFD (*p* = 0.063).

#### 3.3.2. Latency to Enter into the Open Arm

A 2 × 2 × 2 ANOVA was run on the latency (s) to enter into the open arm of the EZM. There was a significant effect of sex, *F*(1, 29) = 13.163, *p* = 0.001, and partial η^2^ = 0.312. Male mice took significantly longer to enter the open arm when compared to female mice.

#### 3.3.3. Open Arm Entries

A 2 × 2 × 2 ANOVA was run on the number of open arm entries that were made in the EZM. There was a significant effect of stress, *F*(1, 29) = 11.005, *p* < 0.01, partial η^2^ = 0.275, sex, *F*(1, 29) = 5.205, *p* < 0.05, partial η^2^ = 0.152, and a significant diet*sex interaction, *F*(1, 29) = 4.976, *p* < 0.05, partial η^2^ = 0.146. Mice experiencing CMUS made significantly greater numbers of open arm entries when compared to NS mice (*p* < 0.01) (Figure 6a) and female mice made more open arm entries than male mice (*p* < 0.05). Female mice given a HFD made significantly less open arm entries compared to females given a CD (*p* < 0.05); male mice on a CD made significantly less open arm entries as compared to females on a CD (*p* < 0.01) (Figure 6b). Females on a CD made a higher number of open arm entries compared to CD males in both NS (*p* < 0.05) and CMUS conditions (*p* = 0.066). In the CD condition, female mice that were given CMUS had a higher average number of entries compared to female mice given NS.

### 3.4. Morris Water Maze (MWM)

#### 3.4.1. Latency to Find Platform

Although mice were able to learn the task over the training days, as seen by a significant effect of day, *F*(5, 145) = 15.159, *p* < 0.001, and partial η^2^ = 0.343 (Figure 7), there were no between-subject effects seen. However, a trending effect of diet*stress was seen, *F*(1, 29) = 3.058, *p* = 0.091, and partial η^2^ = 0.095. For mice receiving no stress, HFD mice took longer to find the platform compared to those on a CD (*p* = 0.095). In those mice that were given a CD, mice given CMUS took longer to find the platform than those in the NS condition (*p* = 0.059).

#### 3.4.2. Thigmotaxicity

Across the training days, the amount of time that mice spent along the outer edge of the pool significantly decreased, *F*(3.403, 98.69) = 66.019, *p* < 0.001, and partial η^2^ = 0.695 (Figure 8). A trending effect of stress was seen, *F*(1, 29) = 4.095, *p* = 0.052, and partial η^2^ = 0.124. Mice undergoing CMUS spent more time in the outer region of the pool than mice under NS.

#### 3.4.3. Day 7 Probe Trial (Percent Time Spent in the Target Quadrant, Day 7)

On the probe trial, there was a significant effect of stress, *F*(1, 29) = 5.347, *p* = 0.028, and partial η^2^ = 0.156, and a significant diet*stress interaction, *F*(1, 29) = 4.451, *p* = 0.044, and partial η^2^ = 0.133. The animals under CMUS spent significantly less time in the target quadrant. Animals on a HFD given CMUS spent significantly less time in the target quadrant on the probe trial as compared to HFD mice given no stress (*p* = 0.004) (Figure 9a). On the probe trial, a significant diet*sex interaction was found, *F*(1, 29) = 11.808, *p* = 0.002, and partial η^2^ = 0.289. (Figure 9b) Simple effects analysis showed that HFD had an effect that was driven by sex. In males, mice that were given a HFD spent significantly less time in the target quadrant when compared to those on a CD, *p* = 0.015. However, in females, mice given a HFD spent significantly more time in the target quadrant compared to those on a CD, *p* = 0.03. Additionally, HFD females receiving NS performed better than those HFD females experiencing CMUS (*p* = 0.001) (Figure 9c).

#### 3.4.4. Day 7 Crosses

On the probe day (day 7), there was a significant effect of stress on the number of platform crosses, *F*(1, 29) = 6.722, *p* = 0.015, and partial η^2^ = 0.188. Mice under CMUS made significantly fewer crosses than those mice with NS (Figure 10).

### 3.5. Nesting

Two raters that were blind to experimental condition rated nests on a scale of 1–5. There was strong agreement between the two raters’ scores for nests: ICC = 0.933 (95% CI: 0.870, 0.965), *p* < 0.001. The two scores were then averaged. Analysis of nesting scores revealed a significant effect of diet, *F*(1, 29) = 5.211, *p* = 0.03, partial η^2^ = 0.152, stress, *F*(1, 29) = 4.648, *p* = 0.04, partial η^2^ = 0.138, sex, *F*(1, 29) = 5.211, *p* = 0.03, and partial η^2^ = 0.152 and a significant stress*sex interaction, *F*(1, 29) = 4.648, *p* = 0.04, and partial η^2^ = 0.138. Mice that were given a HFD built significantly poorer nests than those on a CD (*p* < 0.05) (Figure 11) Mice receiving CMUS built significantly greater nests than those in the NS condition (*p* < 0.05). Males built significantly better nests than females (*p* < 0.05). Female mice that were given CMUS built significantly better nests than females under the NS condition (*p* < 0.01), while males in the NS condition built significantly better nests than females in the NS condition (*p* < 0.01). (Figure 12). Figure 13 shows representative images of nests built by mice in the experimental groups.

### 3.6. Burrowing

A 2 × 2 × 2 ANOVA was run on the amount of pea-gravel burrowed after 2 h. There was a significant effect of stress, *F*(1, 29) = 5.451, *p* < 0.05, and partial η^2^ = 0.158. Mice that underwent CMUS burrowed significantly more pea-gravel when compared to those experiencing NS (*p* < 0.05). (Figure 14) Additionally, there was a significant diet*stress*sex interaction, *F*(1, 29) = 14.093, *p* = 0.001, and partial η^2^ = 0.327. In the NS condition, male mice receiving a HFD burrowed significantly less than those on a CD (*p* < 0.01) and female mice receiving a HFD burrowed significantly more than those on a CD diet (*p* < 0.05). In both dietary conditions, CMUS led to an increase in burrowing, which was further moderated by sex: female mice on a CD experiencing CMUS burrowed more than those NS females on CD (*p* = 0.002) and male mice on a HFD experiencing CMUS burrowed more than those NS males on HFD (*p* = 0.004).

A 2 × 2 × 2 ANOVA revealed no significant differences in the amount burrowed for the overnight measurement.

## 4. Discussion

High-fat diet administration and CMUS during adolescence led to significant impacts on both cognitive and noncognitive behaviors in young adult mice. Mice that were given CMUS displayed increased anxiety in the OFT while displaying less anxiety in the EZM, a pattern of behavior that is thought to be reflective of behavioral disinhibition. Spatial memory deficits were noted in those mice given CMUS and in mice receiving both CMUS and HFD. Apart from cognitive behaviors that are typically assessed in the literature, this paper shows that HFD and CMUS also impact noncognitive behaviors, including nest building and burrowing behavior.

Over the course of the experiment, animal weights increased, with stressed mice weighing significantly less than NS mice during weeks 2–5. Over the four weeks of CMUS administration specifically, CMUS mice weighed less than NS mice in the last three weeks (*p* < 0.05 (week 2) and *p* < 0.01 (weeks 3 and 4)), and they also had differences in the average food consumption. During other CMUS procedures [53,54,55], the stressed mice have been shown to weigh less than non-stressed controls as well as have decreased food intake [55]. However, at the end of the present experiment (week 6), there was no significant weight difference between NS and CMUS mice; this may be because, over the last two weeks of the experiment (weeks 5 and 6, Figure 2a), behavioral testing was taking place and CMUS was no longer being implemented. While studies have shown that stress can lead to reductions in body weight [54,55], Pothion et al. (2004) [53] presents evidence of strain effects on weight during CMUS. Pothion et al. (2004) [53] used a seven-week CMUS paradigm and measured the weights in three separate strains of mice (CBA/H, C57BL/6, and DBA/2). Interestingly, stressed C57BL/6 mice did not have significant differences in body weights when compared to their control counterparts (although they did weigh less). The mice in the present study (C57BL/6J) mice did show reduced weight specifically during the course of the CMUS paradigm. Additional measures, such as sucrose preference to measure anhedonia, may be useful in future studies assessing stress, because CMUS has been shown to significantly reduce sucrose preference in C57BL/6 mice [53].

Stressed mice spent less time in the center of the OF and more time in the open arms of the EZM; this pattern of behavior is thought to be reflective of behavioral disinhibition, a pattern of behavior that has been noted in AD mice [56,57]. However, hyperactivity may be a reason underlying the increase in open arm entries in AD mice, as noted by Gil-Bea et al. (2007) [56]. Stressed mice made significantly greater number of open arm entries compared to NS mice, spent more time in the open arms, and showed more anxiety in the OFT by spending less time in the center. No major differences in distance travelled in the OFT were noted, other than CD males traveling further than CD females.

Bridgewater et al. (2017) [36] noted that, in mice given HFD (the same D12492 diet (Research Diets, Inc.) used in the present study) and chronic stress, males had less locomotor activity than females. In the present study, males on a CD traveled further in the OFT than CD females. While in the EZM, CD males made significantly less open arm entries than CD females. Mice in the present study were given HFD for a period of four weeks prior to testing, while those in [36] were assessed after 12 weeks on the diet. Bridgewater et al. (2017) [36] also conducted these behavioral tests twice, with the males traveling less than females in the OFT occurring prior to introduction of chronic stress. HFD did not affect males in the present study the same way that it did in the [36] study, where male mice on a HFD (after chronic stress) had a reduction in locomotion as compared to those males on a control diet. The control diet that was used in the present study and that used by [36] differed, possibly playing a role in the differences between dietary groups between the studies.

This study utilized both male and female mice, while some researchers have solely focused on one sex in either mice or rats given HFD (females only—[58,59,60], males only—[23,61,62,63]) or chronic stress (males only—[5,34,35,37,38,64,65,66,67]). Even studies that have investigated both chronic stress and poor diet have solely focused on one sex [42]. In female mice, HFD has been shown to impact learning and memory in the 3xTg mouse model of AD [59]. On the probe trial, 3xTg mice given a HFD spent the least amount of time in the target quadrant, similar to the current findings showing that CMUS and HFD mice spent less time in the target quadrant during the probe trial. However, the work done by Sah et al. (2017) [59] was done in the pre-aging stage. Work by Molteni et al. (2002) [60] has also shown that female rats given a high-fat, refined sugar diet had impairments in the MWM when compared to those that were given a regular diet; additionally, these deficits in learning were accompanied by decreases in hippocampal brain-derived neurotrophic factor (BDNF) mRNA and protein.

While most papers have focused on only using a single sex, several papers have assessed both sexes while studying chronic stress [2,4,39] as well as when studying both stress and HFD administration ([36] as discussed above). Unlike the current findings, Naninck et al. (2015) [39] found that chronic early life stress (administered by housing dams and pups in a cage with limited nesting material) did not alter performance in adult mice in the elevated-plus maze (EPM), nor were there significant effects of sex. Here, males took longer to enter into the open arms than female mice and females on a CD spent more time in the open arms than males on a CD. Adolescent rats that were stressed in [2] through social stress by isolation and housing with a new cage partner showed that this stressor had no effect in males on EPM at 45 days of age. Females undergoing this stressor, which were tested at 45 days of age, had less anxiety, as seen in behavior in the EPM, spending more time in the open arms, and having a higher open arm/total arm entry ratio.

In the present study, stressed mice, overall, spent more time in the open arms of the EZM with female mice making more open arm entries when compared to males and having shorter latencies to enter the open arm. Female mice consuming a CD also spent significantly more time in the open arms as compared to males on a CD. This sex difference in anxiety has been reported elsewhere using the EPM [68]. Xiang et al. (2011) [68] showed sex differences in the EPM, with female rats having less anxiety than males. This sex difference was attributed to differences in the AMPA receptor subunit GluR1 in the hippocampus. The total levels of this protein were shown to be lower already in females compared to males and there was a significant relationship between the total levels of GluR1 and time spent in the open arm (indicative of less anxiety).

Bourke et al. (2011) [4] showed that female rats that were exposed to chronic stress during adolescence and tested in adolescence consumed lower amounts of sucrose in the sucrose-preference test (anhedonia) and had a higher number of total arm entries in the EPM when compared to control females. This same pattern was seen when the females were tested at adulthood. Male mice, which were tested in adolescence as well as those tested at adulthood, did not have any significant differences between control and stress conditions. EPM conducted by Bourke et al. (2011) [4] was conducted in the dark phase, which may explain higher levels of EPM activity.

Sex was shown to impact behavior and it interacted with diet in numerous instances. The interaction effects regarding sex in this study should be considered cautiously, as the breakdown by sex resulted in smaller than ideal sample sizes. However, it is important to note from these interactions that both sexes should be included when studying the behavioral effects of HFD and CMUS. For instance, sex was shown to interact with diet in the total distance travelled in the OFT, the number of open arm entries in the EZM, and in the percent time spent in the target quadrant on the probe day in the MWM (Figure 9b). Although these interactions require caution in interpretation, they serve as important reminders that using both sexes of mice in even basic behavioral research is crucial.

CMUS mice made less crosses in the MWM than NS mice, and they also spent less time in the target quadrant on the seventh day probe trial. Additionally, HFD administration, while alone, did not produce significant between-group differences, on the seventh day probe trial it significantly interacted with chronic stress. CMUS HFD mice spent significantly less time in the target quadrant than HFD mice under NS, which indicates that CMUS alongside HFD led to an exacerbated memory impairment than simply HFD alone.

Stressed mice showed impairments in hippocampal memory; this was unsurprising. given that stress has been shown to negatively affect spatial memory in animal models [38,42,64,69,70,71]. CMUS mice made less crosses than NS mice on the 24-h probe day (day 7) and spent less time in the target quadrant on the 24-h probe trial. Additionally, the stressed mice showed a trending effect in thigmotaxicity, spending more time along the edge of the pool than mice under NS. This effect of stress on MWM performance is in agreement with other research on chronic stress and spatial memory [71,72]. There was a significant interaction between diet and stress, with CMUS HFD mice spending significantly less time in the target quadrant than NS HFD mice. In male rats, [42] those receiving both high-fat diet and chronic unpredictable mild stress spent significantly less time than in the target quadrant of the MWM than the control rats, high-fat diet rats, and stressed-alone rats. It is proposed that the combination of HFD and stress resulted in impairments in cognition based on an effect on levels of the leptin receptor (LepRb), with lower levels of LepRb being seen in the hippocampus [42]. Here, the CMUS/HFD mice spent less time in the target quadrant when compared to just those receiving HFD. Yang et al. (2016) [42] only used one sex in their analysis and performed their study in rats; despite this, their findings show agreement as to how these two lifestyle factors may combine to produce even worse behavior through the possible involvement of hippocampal proteins.

HFD administration alone during adolescence has been shown to affect performance in the MWM [73], with juvenile male rats exposed to a HFD containing 24% fat making significantly fewer target crosses than those receiving a control diet. Boitard et al. (2014) [73] went on to show that following an injection of LPS (creating an immune challenge), juvenile rats that had been exposed to a HFD had an increase in inflammatory cytokine activity in the hippocampus. However, there were not any significant differences between juvenile dietary conditions in non-LPS conditions. HFD has also been shown to negatively affect BDNF levels in the hippocampus, accompanying learning and memory deficits in the MWM task in female rats [60], and induce inflammatory responses and insulin resistance in the hippocampi of juvenile male mice given a HFD [74].

Chronic mild stress has been shown to affect long-term potentiation (LTP) in the hippocampus [67], being evident even after a stress paradigm of three days. However, this paradigm did not result in significant differences in spatial memory between the stressed and non-stressed conditions. This may be due to the length of the stressor manipulation, as the present study did show hippocampal-behavioral impairments after four weeks of chronic stress. The effect of stress on the current study’s animal subjects may be due to the numerous effects that stress has on the hippocampus, such as impacting BDNF levels [67,75] or long-term potentiation in CA1 neurons [67].

Daily living measures, representing noncognitive behaviors, are also important to assess. While many studies focus on the cognitive aspects, far fewer include noncognitive activities of daily living measures, which are thought to be a potential measure of well-being [76]. Burrowing and nesting are simple methods to run that provide additional information accompanying changes in cognition. Losses in these noncognitive measures may be early signs of mental decline (e.g., Alzheimer’s disease) [49,77]. For example, mice modeling AD are notoriously poor nest builders when compared to wildtype mice [57,77,78,79] and they show alterations in burrowing activity [57]. Lippi et al. (2018) [57] also show that alterations in ADL measures are seen early on in AD progression.

Stressed mice burrowed significantly more pea-gravel than NS mice and also built better nests than the control mice. In regards to nest construction, other models of stress have shown this to not always be the case. Otabi et al. (2016) [80] used mice that underwent subchronic and mild social defeat stress. In this study, although the mice that underwent this stressor ended up building nests, the onset to build the nests were delayed when compared to the control animals. This was seen again in acute mild social defeat stress done by Otabi et al. (2020) [81], where, although, after 24 h, mice undergoing stress and the control mice had no differences in nest volume, there was delayed nest building in the stressed mice. In the current study, nests were evaluated after 24 h, with significant differences being noted between the groups (Figure 11 and Figure 13). However, whether there is a difference in progressive nest construction, as has been shown in other studies ([77]) between groups is not known and should be investigated in future studies.

Male mice built better nests than females, and this was the case in the NS condition. However, this is not necessarily always seen in the literature, as Xiong et al. (2018) [82] showed. In C57BL/6J mice at seven months vs. 25 months, male mice built worse nests than females at 25 months, but there was no difference at seven months. Additionally, Estep et al. (1975) [83] studied the differences in nest building behavior in wild house mice as compared to C57BL/6J mice, while using both males and females. No sex differences were seen, despite there being strain differences, with C57BL/6J mice building covering nests more than the wild mice and wild mice building no nests more frequently.

The type of nesting material should also be considered, as discussed by Neely et al. (2019) [50]. In the analysis of nest building, researchers may use different materials. Estep et al. (1975) [83] used unbleached cotton batting, Torres-Lista & Giménez-Llort (2013) [77] compared paper towel and cotton, while Xiong et al. (2018) [82] used white paper (similar to the current experiment). Neely et al. (2019) [50] has shown that the type of material used to evaluate nesting can give different responses by raters that are blind to experimental condition, with higher average nest scores being given to mice using shredded paper (as done here and in previous research [57,79]) as compared to cotton squares, bedding, and paper twists. Shredded paper has also been shown to lead to more consistent results [50].

Our finding that HFD negatively impacted nest building is in agreement with previous research studying HFD given to juvenile mice [74]. Vinuesa et al. (2016) [74] used one-month old male C57BL/6J mice and gave them a HFD for four months. They found that, alongside spatial memory deficits, HFD mice built significantly poorer nests.

Additional considerations that should be made include reflecting on the specific type of diet used. Yang et al. (2016) [42] used the D12492 diet from Research Diets, Inc. and found that those undergoing chronic stress and HFD administration had worse spatial memory than control, stress alone, and HFD alone rats. The non-HFD rats in their study did not receive the same control diet that was given in the present study (D12450J); whether or not this difference was the reason why CMUS + HFD were not significantly different from the control conditions (as seen in [42]) is not known. However, utilizing adequate control diets should be considered when manipulating the diet conditions in any study.

D12492 (rodent diet with 60 kcal% fat) has also been used in various studies showing that HFD leads to an increase in anxiety in the OFT [58], impairments in spatial memory [62], increases in blood glucose and insulin when paired together with stress [84], and sex differences with regards to behavioral testing [36]. While similar diets may be used in different studies, additional items to consider include the length of the dietary intervention and when the dietary intervention begins. For instance, Sah et al. (2017) [59] administered HFDs (60% fat) for 16 weeks, starting at week 4, Sasaki et al. (2014) [85] gave HFD (D12492) to pregnant rat moms to assess changes in adolescent offspring, and Zuloaga et al. (2016) [62] placed mice on HFDs (D12492) for three months. In the present study, HFD (D12492) interacted with CMUS, with mice receiving both showing impaired spatial memory when compared to those just receiving a HFD. Additional trending effects were seen in the MWM with the HFD mice taking longer, on average, to find the platform. The effects in the burrowing assay seen in HFD mice were further moderated by sex.

Other types of dietary interventions (poor diets), including cafeteria diet [86,87,88], high-sugar diets [89], or diet manipulations involving adjustments to both fat and sugar [41,60,90,91] should also be considered in future research exploring the effect that diet has on cognitive function and physical health.

Apart from the type of diet given, when assessing effects of stress and diet, the type and amount of stressor given should also be considered. Numerous chronic stress paradigms target a specific type of stressor (immobilization stress—[32], psychosocial/social/social defeat stress—[2,38,64], and limiting nesting/bedding material—[39]), while the present study focused on several different types of stressors in a randomized manner, similar to ([5,33,34,35,37,42,43,44,66,92]). Regarding stress effects, numerous factors can impact the behavioral and cognitive impairments, including: severity of stressors, length of times the stressors are given, and when the stressors are administered. Therefore, future studies should keep these in mind to model real world exposure to more stressful events and how alleviating this stress can lead to better behavioral outcomes.

## 5. Conclusions

In summary, mice given HFD and CMUS during the period of adolescence had alterations in both cognitive and non-cognitive behaviors when tested at young adulthood. Mice under the current CMUS paradigm displayed behavioral disinhibition and alterations in spatial memory. Together, CMUS and HFD affected spatial memory and, apart, affected burrowing and nesting behavior (measures of daily living). The modification of lifestyle factors, including diet and stress during adolescence, serves as a potential strategy to improve cognition in young adulthood.

## Figures and Tables

**Figure 1 brainsci-11-00260-f001:**
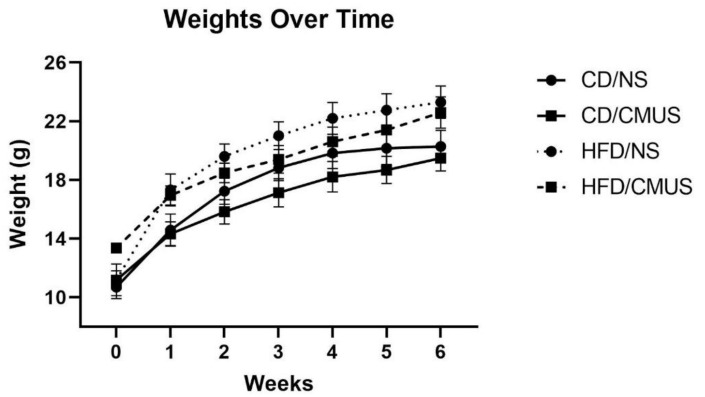
Weights over time. Weights increased significantly over the course of the experiment (*p* < 0.001). High-fat diet (HFD) mice weighed significantly more than those on control diet (CD) (*p* < 0.001) Week 0 represents the start of experimental conditions (baseline—PND 28).

**Figure 2 brainsci-11-00260-f002:**
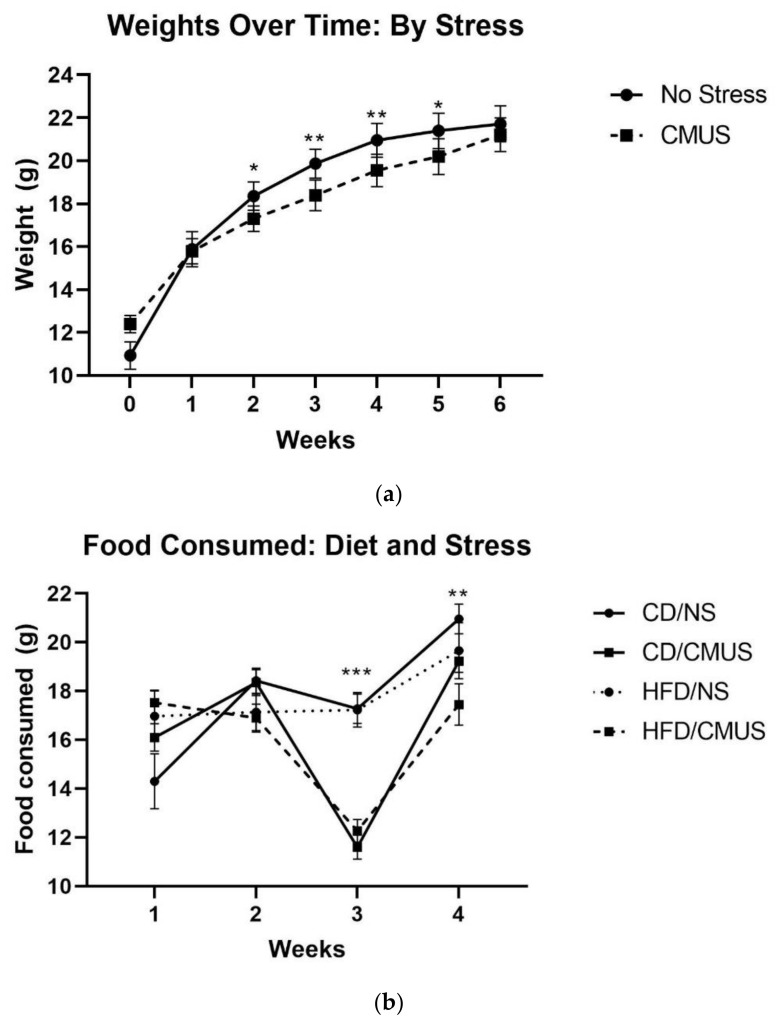
(**a**) Weights over time by stress. Mice experiencing NS weighed significantly more than those in the chronic mild unpredictable stress (CMUS) condition on weeks 2–5 (* *p* < 0.05, ** *p* < 0.01). Week 0 represents the start of experimental conditions (baseline—PND 28). Week 5 and 6 represents the period of behavioral testing (no CMUS occurred during this time). (**b**) Food weights over time by experimental group. CMUS was administered to half of the experimental animals for 4 weeks. Mice experiencing CMUS ate significantly less food than NS mice on weeks 3 and 4 (** *p* < 0.01, *** *p* < 0.001). Due to group housing, food consumed (g) for each animal was calculated as an average.

**Figure 3 brainsci-11-00260-f003:**
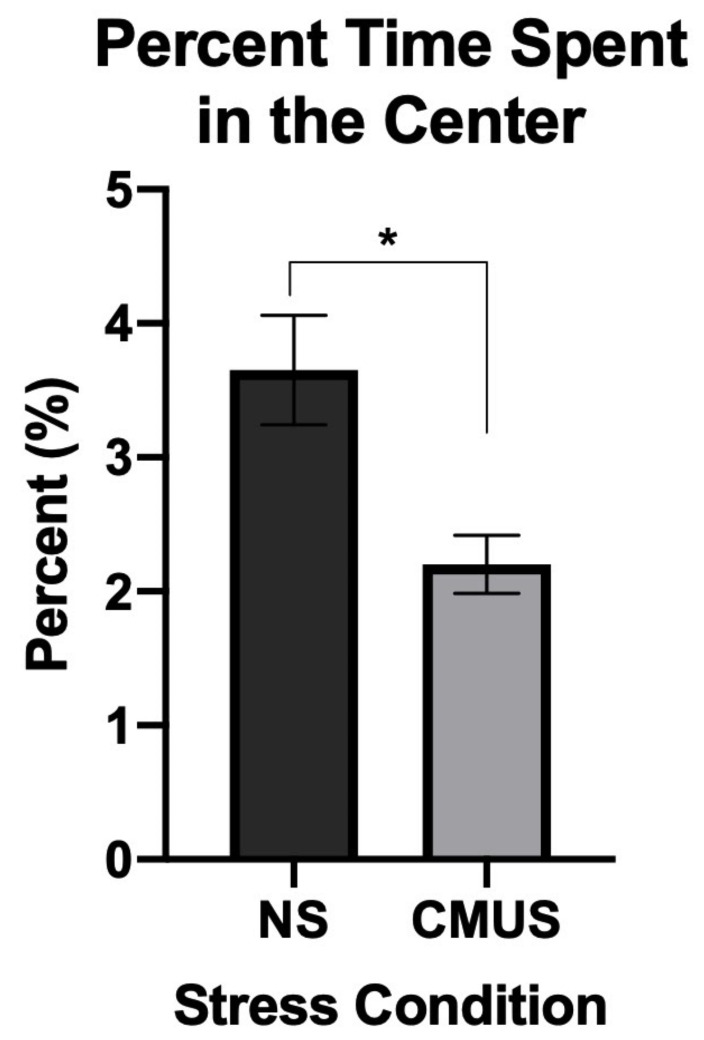
Percent time spent in the center of the open field test (OFT). Mice undergoing CMUS spent significantly less time in the center of the OF compared to NS mice (NS) (* *p* < 0.05; *p* = 0.01).

**Figure 4 brainsci-11-00260-f004:**
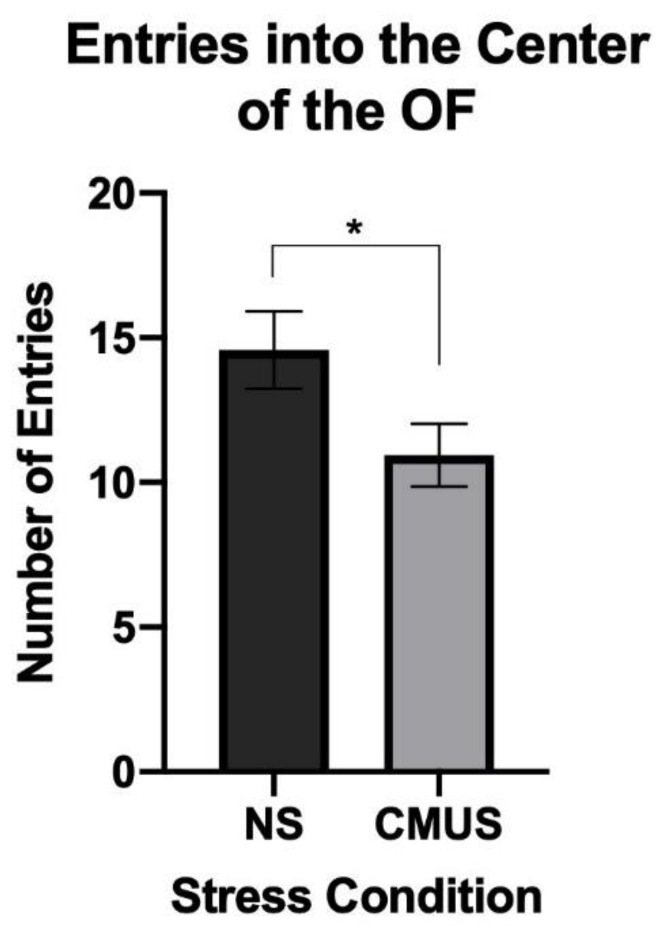
Number of entries into the center of the OFT. Mice undergoing CMUS made significantly fewer entries into the center of the OFT as compared to NS mice (* *p* < 0.05).

**Figure 5 brainsci-11-00260-f005:**
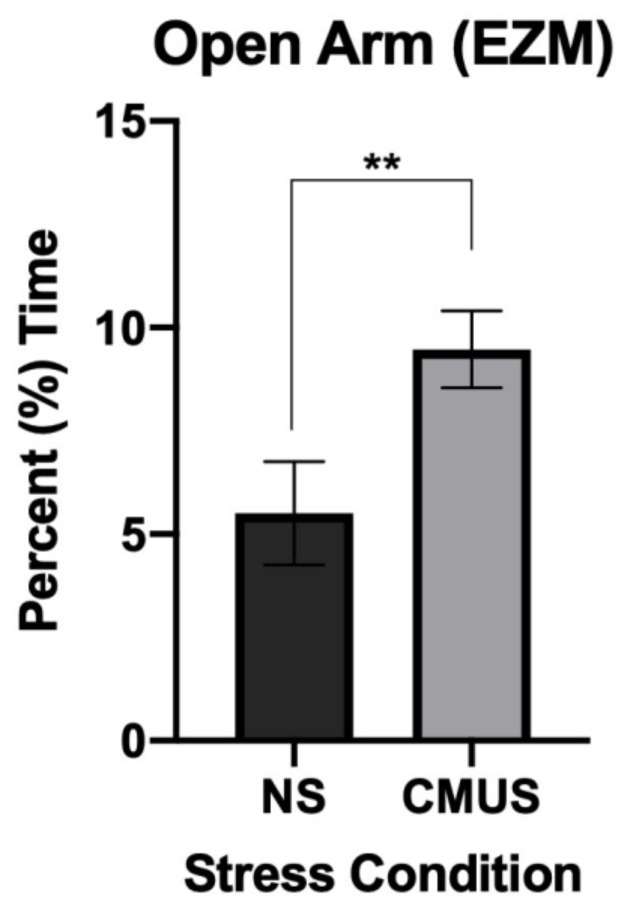
Percent time spent in the open arms of the elevated zero maze (EZM). Mice undergoing CMUS spent significantly more time in the open arms of the EZM compared to those in the NS condition (** *p* < 0.01).

**Figure 6 brainsci-11-00260-f006:**
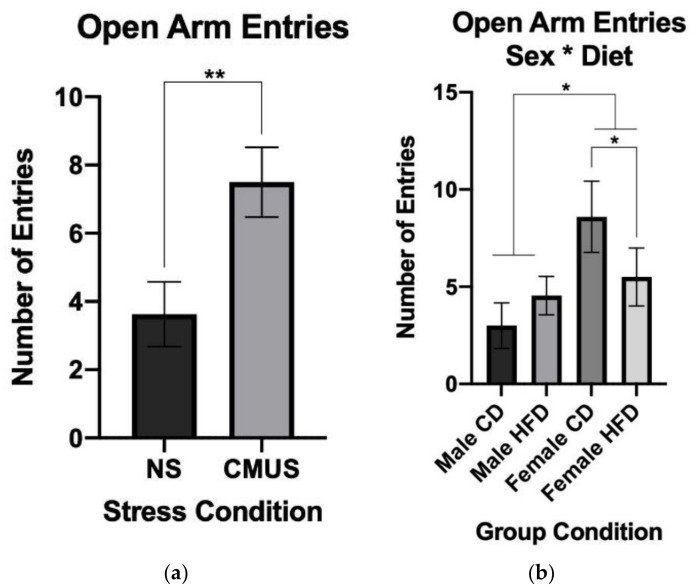
(**a**) Open arm entries in the EZM. Mice in the CMUS condition entered the open arm significantly more than those in the NS condition (** *p* < 0.01). (**b**) Open arm entries in the EZM (sex*diet). Female mice made significantly more entries into the open arm of the EZM than male mice. Female mice given a HFD made significantly fewer open arm entries compared to female mice in the CD condition (* *p* < 0.05). Females on a CD made a higher number of open arm entries compared to CD males in both NS (*p* < 0.05) and CMUS conditions (*p* = 0.066).

**Figure 7 brainsci-11-00260-f007:**
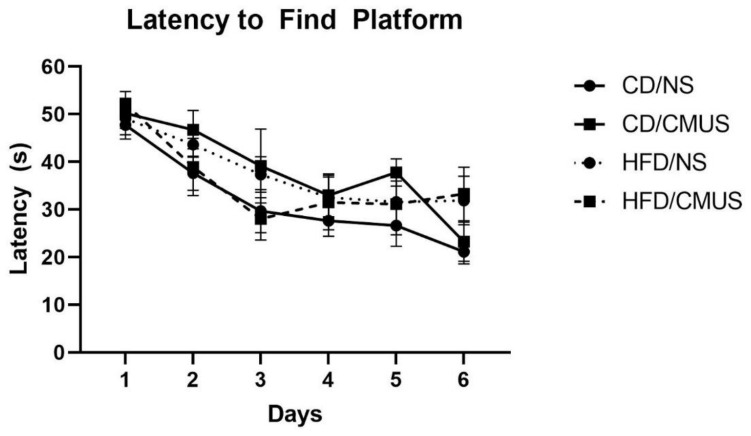
Latency to find the platform. Mice were able to learn to locate the hidden platform in the Morris Water Maze (MWM) task over time (*p* < 0.001).

**Figure 8 brainsci-11-00260-f008:**
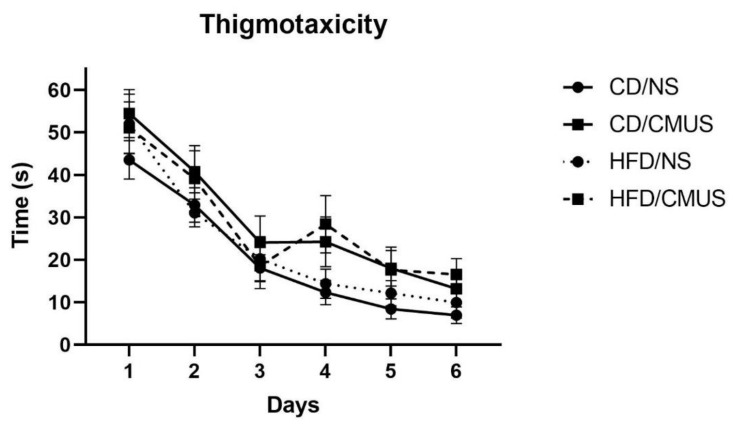
Thigmotaxicity across training days. Across training days, the amount of time mice spent along the border of the MWM significantly decreased (*p* < 0.001).

**Figure 9 brainsci-11-00260-f009:**
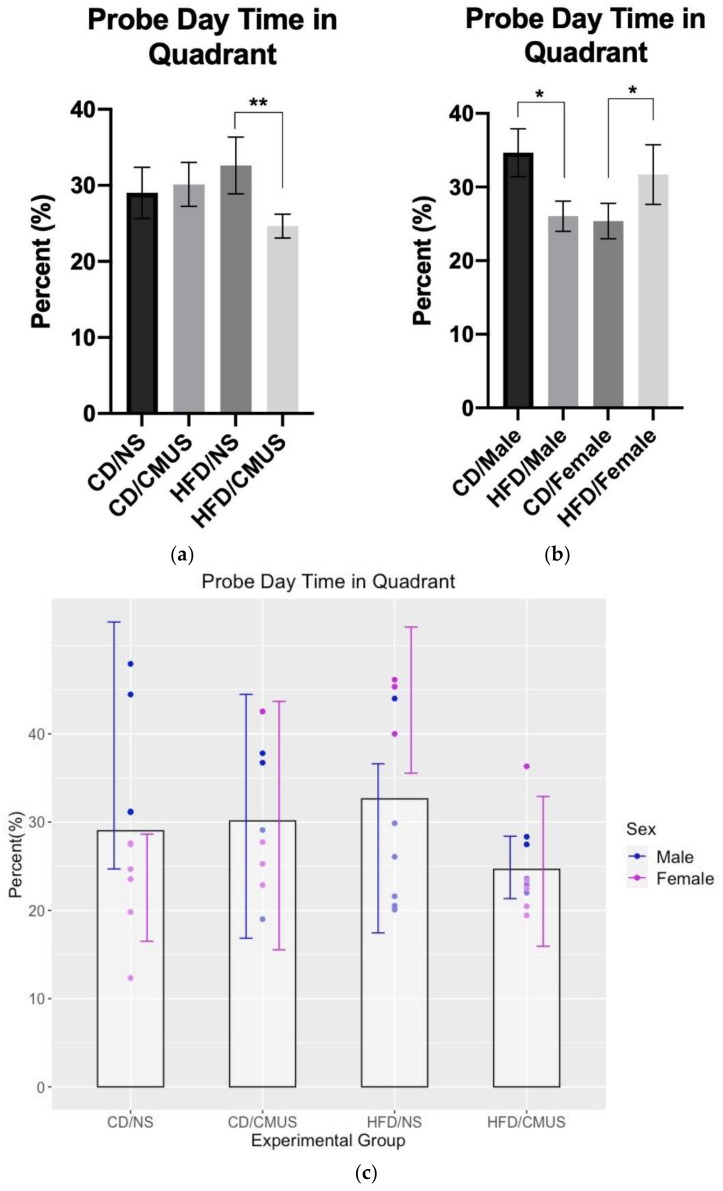
(**a**) Percent time spent in the target quadrant on the probe trial (day 7). Animals experiencing NS spent significantly more time in the target quadrant as compared to those experiencing CMUS (*p* < 0.05). In the high-fat condition, those mice experiencing CMUS spent significantly less time in the target quadrant than those with NS (** *p* < 0.01). (**b**) Percent time spent in the target quadrant on the probe trial (diet*sex). Male mice given a HFD spent significantly less time in the target quadrant than male mice on a CD (*p* = 0.015); female mice given a CD spent significantly less time in the target quadrant than female mice on a HFD diet (*p* = 0.03) (* *p* < 0.05). (**c**) Sex differences in percent time spent in the target quadrant on the probe trial (experimental groups (Figure 9a)). In female mice given HFD, those receiving CMUS spent significantly less time in the target quadrant compared to those given NS (*p* = 0.001).

**Figure 10 brainsci-11-00260-f010:**
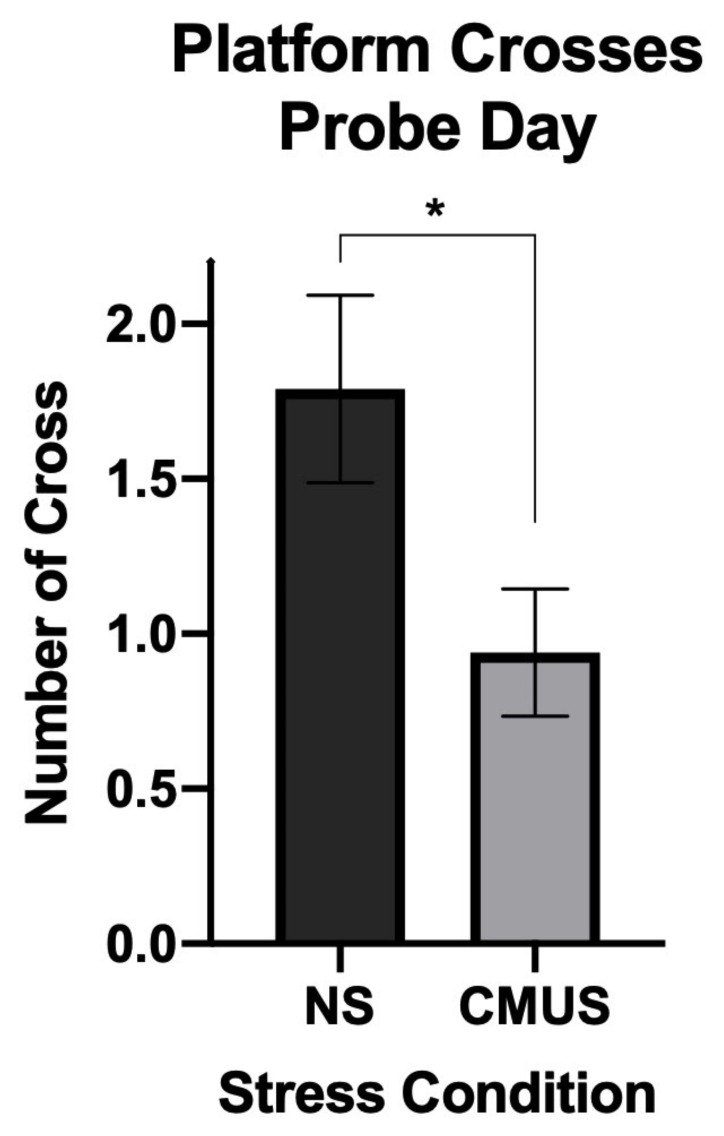
Number of platform crosses on the 24 h Probe Day (Day 7). Mice experiencing CMUS made significantly fewer crosses over the submerged platform as compared to mice experiencing no stress (NS) (* *p* < 0.05).

**Figure 11 brainsci-11-00260-f011:**
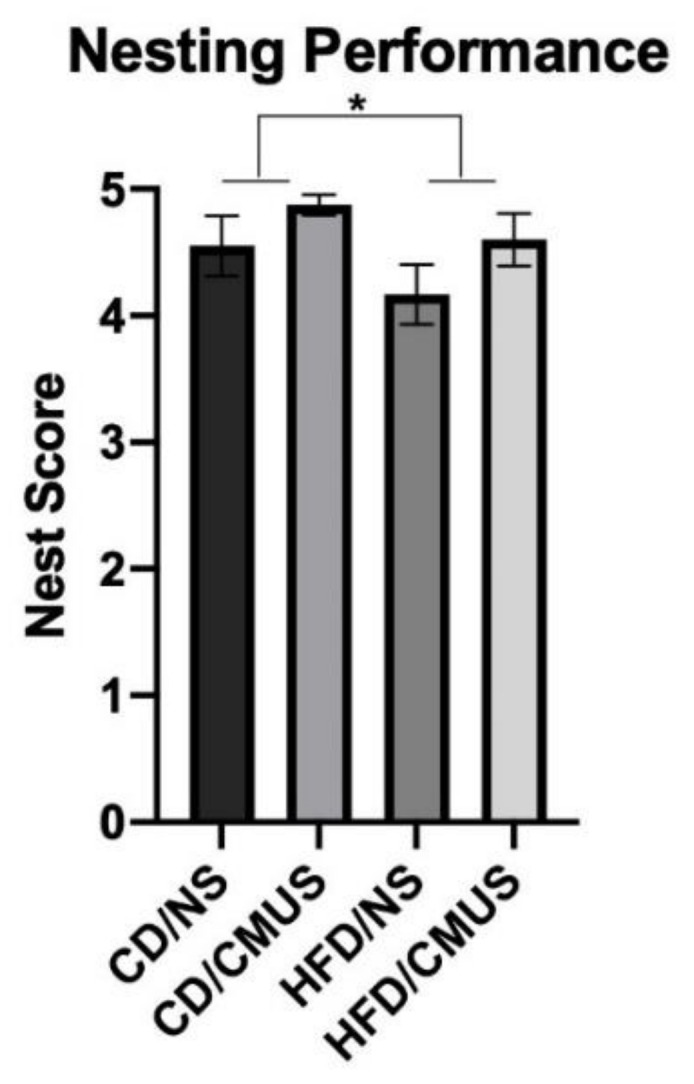
Nesting behavior. Nest construction was significantly altered by HFD administration. Mice given a HFD had significantly lower nest scores as compared to control mice. (* *p* < 0.05).

**Figure 12 brainsci-11-00260-f012:**
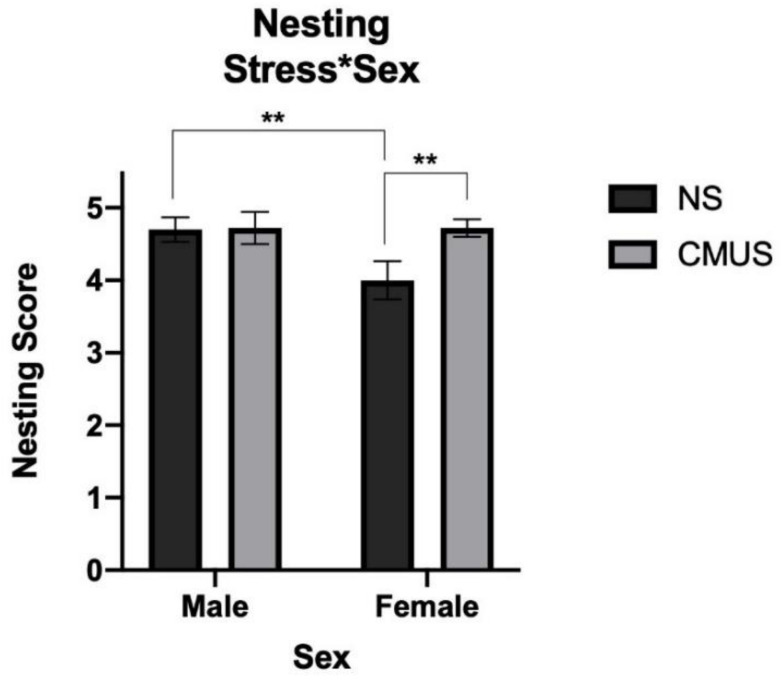
Nesting scores (stress*sex). Female mice undergoing CMUS made significantly better nests than females in the NS condition. Male mice under NS made significantly better nests than females in the NS condition (** *p* < 0.01).

**Figure 13 brainsci-11-00260-f013:**
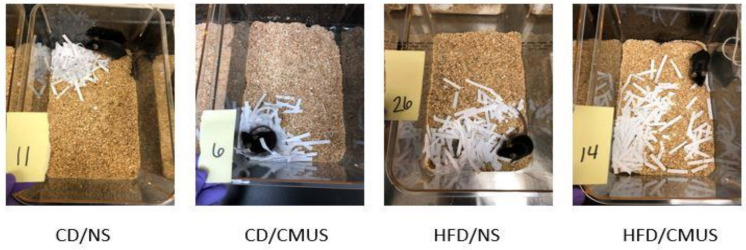
Representative nests built by mice in the experimental conditions. Mice given HFD built significantly worse nests than those given CD (*p* < 0.05). CMUS mice built significantly better nests than those under NS (*p* < 0.05). 11—Control Diet, No Stress; 6—Control Diet, Chronic Mild Unpredictable Stress; 26—High Fat Diet, No Stress; 14—High Fat Diet, Chronic Mild Unpredictable Stress.

**Figure 14 brainsci-11-00260-f014:**
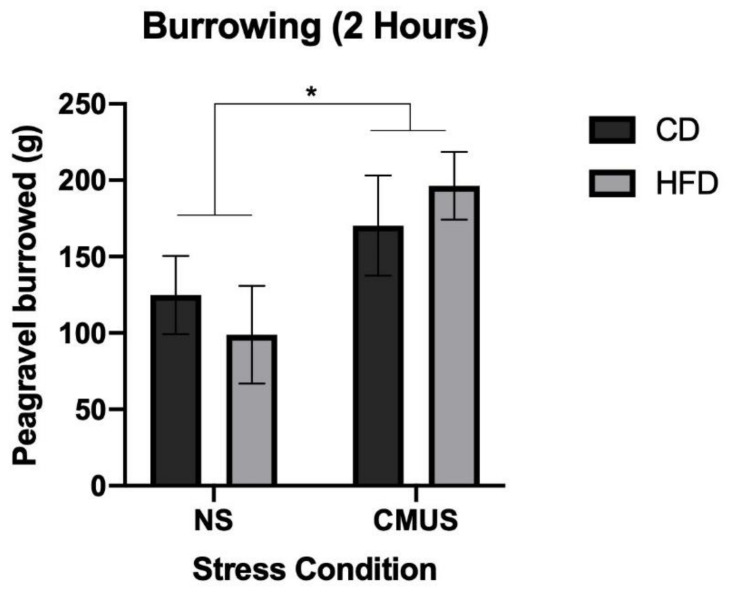
Burrowing behavior after 2 h. CMUS mice burrowed significantly more pea-gravel than NS mice (* *p* < 0.05).

**Table 1 brainsci-11-00260-t001:** Experimental sample size.

	Control Diet (CD)	High-Fat Diet (HFD)	
No Stress	10	9	N = 19
(NS)	(4M, 6F)	(6M, 3F)	
Chronic Mild Unpredictable Stress	8	10	N = 18
(CMUS)	(4M, 4F)	(5M, 5F)	
	N = 18	N = 19	N = 37

**Table 2 brainsci-11-00260-t002:** List of Stressors and their Procedures.

Stressor	Procedure
**Forced Swim**—cold water swim (8–10 °C) for 5 min	Mice are placed into a container filled with cold water (8–10 °C) for 5 min. Water is changed between mice and temperature is checked prior to placing mice into the container. Mice are placed under a heat lamp after completion of the stressor.
**Overnight water deprivation**—during the dark phase (12 h)	Water bottles are removed from the cage before lights out and are replaced at lights on the following morning.
**Bright light****/open field exposure** (10 min)	Mice are placed in an elevated open field box (not the same as the open field box used in the Open Field Test (see Section Open Field Test (OFT))) with 2 overhead lights shining down on them for 10 min.
**Altered Light Cycle** (during dark phase)	During the dark phase, animals in the stress groups are transported into the testing room and are kept in ‘lights on’ for the evening. At the conclusion of the dark phase, animals are placed back onto the caging racks in the vivarium.
**Da****mp bedding** (2 h)	Animal bedding is soaked with water and mice are left in the cage for 2 h.
**No bedding** (2 h)	Animal bedding is taken out and mice are placed into the empty cage for 2 h.
**Overnight social isolation** (12 h)	Animals are housed in individual shoebox cages (Ancare) with access to food and water overnight (12 h).
**Tilted Cage** (1 h)	Animal home cages are tilted for 1 h.
**Predator Urine** (1 h)	A small amount (~1 mL) of bobcat urine is placed on a cotton ball in a centrifuge tube in the mouse’s cage (mice have no direct contact with the urine). This is left in the cage for 1 h. After this stressor, mice are placed into a new home cage.
**Bath (no bedding)** (2 h)	Animal bedding is removed and a small amount of water is placed into the cage, covering the floor. Mice remain in this cage for 2 h.

## Data Availability

The data presented in this study are available on request from the corresponding author, without undue reservation.

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
