# Peer review of "Chronic Mild Unpredictable Stress and High-Fat Diet Given during Adolescence Impact Both Cognitive and Noncognitive Behaviors in Young Adult Mice"

_brainsci, 2021, doi:10.3390/brainsci11020260_

Round 1

Reviewer 1 Report

Overall: In this manuscript by Lippi, they investigate how chronic mild unpredictable stress and high-fat diet interact to modulate anxiety, cognitive, and nesting behaviors in adolescent mice. Overall, mice on the HFD weighed more than CD mice, but there was a notable reduction in gains if mice were also exposed to CMUS. CMUS mice spent less time it the center of the OFT, and had fewer center entries. Interestingly, while males typically traveled further than females, this difference was erased by HFD. For the EZM, females were more likely than males to explore the open arms, though all mice exposed to CMUS showed increased open arm exploration. Regardless of the manipulation, either HFD or CMUS caused an increase in time to find the platform in the MWM. On the probe trial, CMUS decreased time spent in the target quadrant, though female mice on HFD spent more time in the target quadrant. HFD and CMUS had opposite effects on nest building. Overall male mice built better nests than females. HFD had opposite effects on nesting versus burrowing in female mice.

Major Considerations:

  • I am glad the author made a point to examine how the manipulations would affect both male and female mice, but the group sizes are underpowered for any meaningful sex-interaction analysis. I think it would add a valuable dimension to the manuscript if both sexes were powered, but is not necessary for publication. However, any sex-interaction effects should be considered cautiously.
  • The CMUS protocol seems appropriate and varied.
  • Can the author expand on the effect of HFD on locomotion in the OFT? Males traveled more than females when given CD, but this difference was muted if mice had access to the HFD. Is this due to increased locomotion in females, or less locomotion in males?
  • Was the effect of CMUS additive for females on CD in the EZM? Or were they resilient to the effects of CMUS (such that females on HFD, and males on either diet) all rose to meet CD females?
  • What was the lux measurement used for the EZM? Is it possible that high light levels may have driven an opposing anxiolytic effect of CMUS on this assay, whereas low light, which would encourage investigation, may have revealed an anxiogenic effect?
  • HFD had opposite effects on male and female mice in the MWM, but the baselines were also different. Similar to my question above about baseline conditions in the EZM, does the author think that lower target quadrant mice were positively effected by HFD, whereas higher target quadrant mice were negatively effected (which in this case was initially driven by sex?
  • I am a little confused on how to interpret the MWM data in that overall, HFD+CMUS showed a negative effect on time spent in the target quadrant. However, when broken up by sex, the assumption seems to be that HFD females (some of whom were also CMUS?) spent more time in the target quadrant; why is that not reflected then in Figure 9a? I think it would be helpful to display individual data points in these bar graphs with specific signs displaying the other condition (e.g., in the 9a, show data points for male and female mice in each bar graph, but make males one symbol and females a different symbol).
  • I liked the addition of the nesting and burrowing behaviors, and think they present complementary and interesting data.

Author Response

I thank the reviewer for his/her/their review of the manuscript. Per the reviewer's suggestions, I have made the following revisions (highlighted in yellow) in the updated manuscript. 

Figure 9a has been modified. While figure 9a is still being presented, Figure 9c has now been added, illustrating sex differences in the probe day time in target quadrant (Lines 336-340). Figure 9c has been added to Day 7 Probe Trial (3.4.3) with the following added: "Additionally, HFD females receiving NS performed better than those HFD females experiencing CMUS (p = 0.001)" (Lines 324-325). This graph was produced using the ggplot2 package in R. Additional reference to R and this package has been added in the statistical analysis section. (Line 179-180) 

Additional information has now been added to Total Distance in the OFT (3.2.4) (Lines 244-249). A main effect of sex was seen with males travelling further than females (p < .01). Although males on a CD traveled further than females on a CD, males and females ran similar distances in the HFD condition. 

Additional information has now been added to Percent time in the open arm (3.3.1) (Lines 257-260). The following was added: "Female mice consuming a CD spent significantly more time in the open arms compared to males on a CD (p = 0.01). There was also a trend noted for female mice: those consuming a CD spent more time in the open arms compared to females on a HFD (p = 0.063). 

Additional information has now been added to Open arm entries (3.3.3) (Line 278-280). The following has been added: "Females on a CD made a higher number of open arm entries compared to CD males in both NS (p < 0.05) and CMUS conditions (p  = 0.066). In the CD condition, female mice given CMUS had a higher average number of entries compared to female mice given NS. 
Figure 6b's caption has also been edited (Lines 290-291), adding that "Females on a CD made a higher number of open arm entries compared to CD males in both NS (p < 005) and CMUS conditions (p = 0.066). 

Line 428-429 has a clarifying statement added: "..., which indicates that CMUS alongside HFD led to a worse outcome than simply HFD alone." This was made in response to the comment on how to interpret the MWM data in that HFD+CMUS showed a negative effect on time in target quadrant. Compared to HFD alone, the combination of CMUS with HFD led to even worse behavior. 

Lines 467-473 have been added addressing that interaction effects regarding sex should be considered cautiously. It has also been added that, despite the breakdown by sex resulting in smaller than ideal sample sizes, both sexes should be included when studying behavioral effects of HFD and CMUS in future studies. 

Additional discussion on sex differences in behavior has been added (Lines 454-460). 

The lux measurement used for EZM was not recorded. The set up for EZM was such that the overhead laboratory lights were turned off and lamps were spread equidistant around the EZM; this created a less bright space as bright light can have anxiogenic effects in laboratory animals. 

In the EZM, females on a CD had made higher numbers of open arm entries compared to CD males under NS (significant) and CD males under CMUS (trending). Females given a CD under CMUS made a higher average number of entries compared to female mice on a CD under NS. Therefore, the females on a CD seemed to be somewhat resilient (in this measure) to the effects of CMUS. 

In the MWM, although the diet*sex interaction should be viewed cautiously, it is interesting to note that HFD in males led to reduced time spent in the TQ while HFD led females to spend greater amounts of time. However, as seen in Figure 9a, HFD combined with CMUS led to significantly worse memory in the probe trial than HFD alone (& there was no significant difference between HFD alone and CD alone or CD with CMUS). 

Additional changes that have been made to the overall document include grammar changes as well as the addition of references 79-92. 

Reviewer 2 Report

Major comments:

  1. Did the author clean the cage and maze in open field test and elevated-zero maze between each trial? The flavors from the last animal could influence the behavioral response of next one.
  2. CMUS paradigm is a classic model for depression. Did the author have any behavioral results for depressive-like symptoms? CMUS tends to induce anhedonic-like behaviors, did the author measure the food intake during CUMS procedure for each group? 
  3. Could the author explain why stressed and control mice have no difference in body weight at the end of CUMS procedure (week 6)? This result is contradicted with many previous studies. 
  4. This study is pure descriptive, what is the possible mechanism underlying the CUMS and high-fat diet induced behavioral deficit?

Author Response

I want to thank the reviewer for his/her/their review of the manuscript. Per the reviewer's comments, the following changes have been made to the manuscript (as seen by yellow highlighted text):

Yes, the behavioral apparatuses (OFT and EZM) were cleaned between each trial to minimize odor cues between mice with the use of 70% ethanol. This has been added in the methods section of each of these tests (Lines 130-131; Lines 141-142). 

I did not measure any behavioral results for depressive-like symptoms (sucrose preference, forced swim). However, as the reviewer noted, as CMUS can induce anhedonic-like behaviors, food intake was measured. This has been added in the methods section under section 2.1.2 Diets. Due to group housing, average intake per mouse was calculated as (g) food consumed after 1 week / Number of animals in the cage. (Lines 105-107) The results section (3.1 Animal weights and food consumed - lines 186-203) has been updated to reflect the inclusion of analysis of food consumption data. Additional information has been added in Figure 2a (line 214) and a new figure, Figure 2b and its caption has been added (lines 217-220). 

Additional information has been added in the discussion section (section 4) discussing body weights (lines 405-418)

Throughout the discussion section, information has been added that discusses potential brain mechanisms for the behavioral deficits (factors noted in the brain such as inflammation, BDNF expression, and changes in protein levels). 

Additional changes in the document consist of added references (79-92) and changes to grammar when encountered. 

Round 2

Reviewer 2 Report

The authors fully addressed my concerns.